# In Vivo Study of Inoculation Approaches and Pathogenicity in African Swine Fever

**DOI:** 10.3390/vetsci11090403

**Published:** 2024-09-01

**Authors:** Qian Xu, Dongfan Li, Xiaoyu Chen, Xiaoli Liu, Hua Cao, Hui Wang, Haowei Wu, Tangyu Cheng, Wenhui Ren, Fengqin Xu, Qigai He, Xuexiang Yu, Wentao Li

**Affiliations:** 1College of Veterinary Medicine, Huazhong Agricultural University, Wuhan 430070, China; xq1@webmail.hzau.edu.cn (Q.X.); lidongfan@webmail.hzau.edu.cn (D.L.); chenxiaoyv@webmail.hzau.edu.cn (X.C.); xiaoliliu@mail.hzau.edu.cn (X.L.); caohua@webmail.hzau.edu.cn (H.C.); Wang0158@webmail.hzau.edu.cn (H.W.); wuhaowei0701@163.com (H.W.); chengtangyu@mail.hzau.edu.cn (T.C.); renwenhui8962@163.com (W.R.); xufengqin2005@163.com (F.X.); he628@mail.hzau.edu.cn (Q.H.); 2National Key Laboratory of Agricultural Microbiology, Wuhan 430070, China; 3The Cooperative Innovation Center for Sustainable Pig Production, Wuhan 430070, China; 4Key Laboratory of Prevention & Control for African Swine Fever and Other Major Pig Diseases, Ministry of Agriculture and Rural Affairs, Wuhan 430070, China; 5Hubei Hongshan Laboratory, Wuhan 430070, China

**Keywords:** African swine fever virus, pathogenicity, intramuscular inoculation, oronasal inoculation, histopathology

## Abstract

**Simple Summary:**

In this study, pigs were inoculated with the ASFV strain HB31A via intramuscular and oronasal routes to investigate the viral dynamics and pathogenicity of ASFV. Pigs in the intramuscular group exhibited more reproducible clinical disease with a high consistency in disease outcomes. In contrast, pigs in the oronasal group had a longer incubation period, a prolonged course of disease, and a delayed onset of viral shedding. Additionally, one pig in the oronasal group that survived the infection exhibited chronic disease and persistent infection, intermittently excreting the virus and developing low-level viremia. This phenomenon indicates that pigs infected with ASFV in the field can develop non-lethal chronic disease and persistent infection, with intermittent viral excretion, even when infected with a highly virulent strain. These findings suggest that the survival of ASFV-infected pigs in the field poses a significant threat to uninfected pigs and can lead to severe economic losses in the pig industry.

**Abstract:**

African swine fever is an extremely infectious viral disease that can cause nearly 100% mortality in domestic pigs. In this study, we isolated an ASFV strain HB31A and characterized it using hemadsorption assay, immunofluorescence assay, and electron microscopy. We then performed animal experiments on 20-day-old pigs through intramuscular and oronasal inoculations with HB31A. Pigs in the intramuscular group exhibited more consistent clinical disease, with an incubation period of 4.33 ± 0.47 days and a 100% mortality rate within 6.67 (±0.47) days post-inoculation (dpi). In contrast, the oronasal group experienced a longer course of disease, with an incubation period of 6.00 ± 0.82 days. Two out of three pigs in the oronasal group died at 8 and 10 dpi, while the surviving pig exhibited chronic disease and persistent infection, intermittently excreting ASFV through the oral, nasal, and rectal pathways. Virus DNA was found in oral, nasal, and rectal swabs at 1–3 dpi in the intramuscular group and at 3–5 dpi in the oronasal group. In summary, HB31A is highly lethal to domestic pigs, and field-infected pigs have the potential to develop non-lethal, chronic disease and persistent infection, with intermittent viral shedding, even when infected with a highly virulent strain. These findings offer a valuable understanding of the viral dynamics and pathogenicity of ASFV and highlight the difficulties in diagnosing, preventing, and controlling African swine fever.

## 1. Introduction

African swine fever (ASF) is a highly contagious and deadly swine disease that can affect both domestic and wild pigs. ASF is caused by African swine fever virus (ASFV), which is the sole representative of the Asfarviridae family and the only identified DNA arbovirus to date [1]. ASFV’s genome consists of a linear double-stranded DNA molecule, ranging from roughly 170 to 193 kb in length among various isolates [2,3,4]. It encodes 151 to 167 open reading frames (ORFs) [5]. ASFV strains with various genome sizes or compositions are reportedly virulent and low-virulent, causing a spectrum of infections, including peracute, acute, subacute, chronic, or subclinical, with corresponding clinical signs [6,7,8,9]. ASF has been present in sub-Saharan Africa since the 1920s [10]. Since 2007, it has spread to domestic and wild pig populations in other parts of the world. Following the initial report of ASF in China in August 2018, the virus had disseminated to all provinces by mid-2019 [11].

Extensive ASF pathogenicity investigations have depended on the intramuscular (IM) inoculation route [8,12,13,14,15,16,17,18,19], yielding consistently reproducible clinical outcomes. Nevertheless, intramuscular inoculation circumvents innate immune defenses, such as the interaction between the virus and the mucosal surfaces of the mouth and upper respiratory tract [20]. Therefore, this approach may not be the best inoculation route for studying the early pathogenesis and previremic phases of disease, nor should it be used for vaccine evaluation. In addition, it is generally acknowledged that, in domestic pigs under natural conditions, ASFV is mainly transmitted through direct contact with excreted viral particles via ingestion and/or nuzzling [21,22]. Therefore, the oronasal inoculation route may simulate natural infection.

An ASFV strain, HB31A (GenBank Accession number: ON380540), was obtained from a pig serum sample during ASFV field surveillance. Its whole-genome sequence was analyzed in a previous study and clustered as a genotype II ASFV strain [23]. The strain was further characterized by hemadsorption (HAD) assay, immunofluorescence assay, and electron microscopy. To better understand the viral dynamics and pathogenicity of ASFV via either intramuscular or oronasal inoculations, HB31A was used to inoculate pigs through both routes. The outcomes were measured in terms of pathogenicity, survival rate, incubation periods, clinical signs, viral shedding, viremia, antibody responses, viral loads in various tissues, gross lesions, and histopathological changes. 

## 2. Materials and Methods 

### 2.1. Cell Culture and Virus Isolation

PAMs were obtained from the bronchoalveolar lavage of 20–30-day-old pigs free from PRRSV, PCV2, PRV, or CSFV infections, as previously described [24], and the cells were cultured in 10% FBS RPMI 1640 medium (Thermo Scientific, Waltham, MA, USA) at 37 °C with 5% CO_2_. To isolate the virus, a pig serum sample that tested positive for ASFV via qPCR and HAD assay was used to inoculate the PAMs. Supernatants were harvested at 4 dpi, and viral loads were quantified using qPCR for viral gene copies and HAD assay for infectious particles. The aliquoted virus stocks were subsequently stored at −80 °C. 

### 2.2. qPCR

ASFV genomic DNA was extracted from the pig serum sample, cell supernatants, swabs, EDTA-treated whole peripheral blood, and tissue homogenates using TIANamp Genomic DNA Kits. qPCR was carried out on a CFX Connect Real-Time PCR Detection System (Bio-Rad Laboratories, Inc., Shanghai, China) following the procedure recommended by the WOAH. The forward primer used was 5′-CTG-CTC-ATG-GTA-TCA-ATC-TTA-TCG-A-3′, the reverse primer was 5′-GAT-ACC-ACA-AGA-TC(AG)-GCC-GT-3′, and the probe was 5′-FAM-CCA-CGG-GAG-GAA-TAC-CAA-CCC-AGT-G-3′ TAMRA [25]. The complete procedure is provided in Appendix A. 

### 2.3. Hemadsorption Assay

Hemadsorption (HAD) assay was conducted with slight modifications to the previously described method [26]. PAMs were cultured in 96-well plates and infected with 10-fold serial dilutions of the pig serum sample. The amount of ASFV was assessed by detecting rosette characteristic formations, which signify the hemadsorption of erythrocytes around infected PAMs. HAD was monitored microscopically for at least seven days. 

### 2.4. Immunofluorescence Assay

PAMs were cultured in 96-well plates and infected with different doses of ASFVs. Viral replication was subsequently confirmed by immunofluorescence assay (IFA) using anti-ASFV p30 monoclonal antibodies following the methodology previously reported [27].

### 2.5. Electron Microscopy 

PAMs were cultured in T75 cell culture flasks and infected with ASFV at a multiplicity of infection (MOI) of 0.1. At 48 h post-infection (p.i.), cell supernatants were harvested and fixed with 2.5% glutaraldehyde (pH 7.2) for negative staining. A 20 μL aliquot of the sample was applied to a carbon-coated grid that had been glow-discharged and then negatively stained with 2% phosphotungstic acid. In addition, ASFV-infected cells were washed with PBS, fixed with 2.5% glutaraldehyde (pH 7.2) at 4 °C overnight, postfixed with 1% OsO4 (pH 7.4) at 4 °C for 2 h, dehydrated in stepwise acetone at 4 °C, and embedded in 812 Epon resin. The thin sections were stained with 1% uranyl acetate (pH 6.5) and 1% lead citrate (pH 7.2) and then examined using an H7650 electron microscope (Hitachi, Tokyo, Japan) at 80 kV. 

### 2.6. Virus Growth Titration

PAMs were cultured in 6-well plates and infected with the ASFV strain at an MOI of 0.1. Cell cultures were collected at different time points post-infection. To ensure sufficient release of the virus from the cell debris, the cultures were subjected to three cycles of freezing at −80 °C and thawing at 4 °C. The cell debris was then removed by centrifugation, and the supernatant was used for virus growth titration. Viral genomic DNA was extracted from the supernatants, and the copy numbers of the viral B646L gene (p72) were quantified using qPCR. Simultaneously, the 50% tissue culture infectious dose (TCID_50_) was determined by IFA and calculated according to the Reed and Muench method [28]. Three independent experiments were conducted.

### 2.7. Animal Experiments

A group of 20-day-old Large White pigs free from PRRSV, PCV2, PRV, or CSFV infections were randomly divided into three groups, each consisting of three pigs. The pigs in the two experimental groups were inoculated with HB31A at a dose of 200 TCID_50_ (in 200 μL of cell culture medium) via intramuscular inoculation (IM) or oronasal inoculation (ON). Specifically, pigs in Group ON were each inoculated with 100 TCID_50_ orally and nasally. 

The temperature and clinical signs of the pigs were monitored daily for 28 dpi. The clinical scoring scale was designed based on a previously described ASF clinical scoring table (see Appendix A) [29].Oral, nasal, and rectal swabs were obtained daily for viral detection by qPCR. Blood samples were collected at 0, 2, 4, 7, 10, 14, 17, 21, 24, and 28 dpi. Simultaneously, serum samples were collected and assessed for IgG targeting the viral p30 protein, as previously described [27]. If an animal succumbed during the study, a necropsy was performed immediately. Surviving pigs were euthanized and necropsied at the end of the observation period. The histopathologic lesion scoring scale was designed based on a previously described ASF histopathologic lesion scoring table (see Appendix A) [20]. Tissue samples, including from the heart, liver, spleen, lung, kidney, tonsil, multiple lymph nodes (inguinal, hepatic, and mesenteric), stomach, intestines (duodenum, jejunum, ileum, cecum, colon, and rectum), brain (cerebrum, cerebellum, pons, and medulla oblongata), and pancreas, were collected for virus detection and histological analysis.

### 2.8. Histopathological and Immunohistochemical Analysis

The histopathological and immunohistochemical analyses were carried out by Wuhan Baiqiandu Biotechnology Co., Ltd. The tissue samples collected from inoculated pigs were fixed in 10% buffered formalin at room temperature for 48 h. After paraffin embedding, the tissues were sectioned into 4 μm thick slices. Each slide was stained with hematoxylin and eosin, and histological changes were examined using light microscopy.

The ASFV viral antigen in the tissue samples was examined by immunohistochemical analysis. Immunohistochemistry was performed as previously described with minor modifications [30]. Briefly, the sections were deparaffinized, rehydrated, heated in sodium citrate antigen retrieval solution (pH 6.0), and treated with 3% hydrogen peroxide. Nonspecific binding sites were blocked with 3% BSA. Then, the sections were incubated with a primary monoclonal antibody against the viral protein p30 (10 g/mL in PBS) overnight at 4 °C. Following the primary incubation, the sections were washed with PBS (pH 7.4) for 5 min and incubated with a HRP-conjugated secondary antibody (SeraCare Life Sciences, Gaithersburg, MD, USA) at room temperature for 50 min. After staining the sections with 3,3’-diaminobenzidine as the chromogen and counterstaining with hematoxylin, they were dehydrated, cleared in xylene for 5 min, and finally mounted using a resin medium. 

## 3. Results

### 3.1. Isolation and Characterization of HB31A In Vitro

The pig serum sample was collected during ASFV field surveillance. We confirmed that the serum sample was positive for ASFV by qPCR targeting the viral B646L gene, as recommended by WOAH [25]. At 2 dpi with the serum supernatant, the PAMs exhibited pronounced HAD (Figure 1A). Cell supernatants were collected at 4 dpi, creating the first passage stock. The expression of ASFV p30 proteins was detected by IFA at 48 h p.i. in PAMs infected with the virus stock (Figure 1A). Electron microscopy (EM) revealed the characteristic morphology of ASFV particles in the infected PAMs (Figure 1A).

For evaluating the virus’s growth kinetics, PAMs were infected at an MOI of 0.1. Cell cultures were harvested at various intervals post-infection and assessed for viral titers by the TCID_50_ assay and B646L gene quantification by qPCR. The viral titers and B646L gene copies reached 1 × 10^7.12^ TCID_50_/0.1 mL (Figure 1B) and × 10^6.86^/μL at 84 h p.i. (Figure 1C), respectively. These findings confirmed the successful isolation of ASFV from the pig serum sample and its efficient replication in PAMs. The virus isolate was designated as HB31A.

### 3.2. The Pathogenicity of HB31A in Pigs via Intramuscular and Oronasal Inoculations

To characterize the pathogenicity of HB31A and investigate disease signs and viral shedding via intramuscular and oronasal inoculations, three pigs were inoculated intramuscularly with HB31A at a dose of 200 TCID_50_ (Group IM) and three other pigs were inoculated oronasally with the same dose (Group ON). The whole experimental procedure was performed as shown in Figure 2A. The disease signs are shown in Figure 3 andsummarized in Table 1.

All pigs inoculated with HB31A exhibited disease signs, including fever, inappetence, and lethargy. Moreover, the pigs in Group IM showed clinical symptoms earlier. Less consistent clinical signs included wheezing/coughing, nosebleeds, papules, cutaneous necrosis, diarrhea, and archorrhagia. As the disease progressed, the clinical symptoms of pigs in the IM and ON groups that died from infection continued to worsen (Figure 2B,C). In the later stages of ASF, the pigs exhibited symptoms such as anorexia and inability to stand. These pigs all showed acute infection and clinical signs. The incubation period of Group IM (4.33 ± 0.47 days) was shorter than that of Group ON (6.00 ± 0.82 days), but the difference was not significant. The intramuscular inoculation route resulted in the most consistent progression of disease. In Group IM, all pigs developed diarrhea after 4 dpi, two pigs developed papules in the abdominal skin after 4 dpi, one pig developed cutaneous necrosis after 5 dpi, and all pigs died between 6 and 7 dpi. Oronasal inoculation caused later disease signs and a longer course of disease. In Group ON, two pigs died at 8 and 10 dpi, respectively. Between the two pigs, one developed archorrhagia at 8 dpi and the other developed nosebleeds after 9 dpi. Notably, one pig (ON-1) survived the infection during the observation period and exhibited non-lethal, chronic disease and persistent infection. The body temperature of ON-1 exceeded 40 °C at 5 dpi and showed intermittent fevers of varying degrees in the early stages (Figure 2D). Additionally, ON-1 experienced diarrhea from 4 to 8 dpi and developed cutaneous necrosis from 8 to 11 dpi. The pig began coughing at 7 dpi, with symptoms worsening between 15 and 20 dpi. During this period, its appetite and overall condition were at their lowest. By the end of the observation period, ON-1 continued to exhibit coughing symptoms.

### 3.3. Viral Shedding of HB31A in Inoculated Pigs

To assess viral shedding in pigs infected with HB31A, oral, nasal, and rectal swabs were collected to quantify the number of viral DNA copies via qPCR.

In Group IM, the viral shedding time for oral swabs was approximately 3.00 (±0.82) days and about 3.33 (±0.47) days for rectal swabs, both significantly shorter compared to Group ON. However, the viral shedding time for nasal swabs was around 2.33 (±1.25) days, showing no significant difference from Group ON. The maximum viral load titers in oral swabs of Group IM reached approximately 10^6.36 (±0.20)^ copies/mL, which was significantly higher than in Group ON. In nasal and rectal swabs, the maximum viral load titers in Group IM reached around 10^7.20 (±0.50)^ copies/mL and 10^6.80 (±0.17)^ copies/mL, respectively, with no significant difference from Group ON (see Table 2).

Notably, during the observation period, intermittent viral shedding via oral, nasal, and rectal routes was observed in ON-1 (Figure 4). Viral DNA was detectable in oral swabs, except at 10, 20, 21, 22, 23, 27, and 28 dpi (Figure 4A); in nasal swabs, except at 5, 21, 23, 26, 27, and 28 dpi (Figure 4B); and in rectal swabs, except at 19, 20, 21, 23, and 28 dpi (Figure 4C).

Overall, in Group IM, viral loads were higher and detected earlier in swabs compared to Group ON. In both Group IM and Group ON, viral shedding occurred earliest in nasal swabs, followed by oral swabs and rectal swabs.

### 3.4. Viremia and ASFV-Specific Antibody Responses in Inoculated Pigs

To evaluate viremia in pigs infected with HB31A, and to assess the antibody response in the IM and ON groups, viral DNA copies were quantified in collected blood samples. Additionally, serum samples from inoculated pigs were tested for IgG against the ASFV p30 protein using an indirect ELISA.

Viral DNA was detectable in the blood after 2 dpi in Group IM and after 4 dpi in Group ON (Figure 5A). The maximum viral load titers in the blood of Group IM reached approximately 10^9.36 (±0.19)^ copies/mL, showing a very significant difference compared to Group ON.

Viral DNA was also detectable in the blood of ON-1 at the indicated time points after 7 dpi, which developed low-level viremia (Figure 5A). Additionally, between the two groups, only ON-1 seroconverted at 14 dpi (Figure 5B).

These results indicate that viremia occurred later in Group ON compared to Group IM and that the pigs with acute death failed to produce antibody responses to ASFV.

### 3.5. Lesions and Virus Replication in Tissues

Tissue samples were obtained for quantified viral DNA copies by qPCR from necropsied pigs on the day of their death and from ON-1, which was euthanized at 28 dpi. A summary of gross lesions in all pigs is provided in Appendix A, the gross lesions of IM-2, along with ON-1 and ON-2 in Group ON, are illustrated in Figure 6A. IM-2 and ON-2 exhibited the most severe gross lesions within their respective groups.

The histopathologic lesion scores of all pigs in Group IM were higher than those in Group ON, but there was no significant difference between the two groups (Figure 6B,C). Five pigs from the two groups that died from ASFV infection exhibited typical pathological changes associated with African swine fever upon necropsy, although the severity of the lesions varied. Common pathological changes observed in all pigs included hepatomegaly, splenic and lymph node swelling/hyperemia, petechiae in the renal cortex, and renal medullary hemorrhage. The heart predominantly displayed hemorrhagic spots (4/5 pigs), while the lungs showed edema (4/5 pigs), hyperemia (4/5 pigs), petechiae (1/5 pigs), and consolidation (3/5 pigs). The gastrointestinal tract exhibited hemorrhage (4/5 pigs), and the gallbladder showed edema (4/5 pigs) and hemorrhage (2/5 pigs). In contrast, ON-1, in addition to displaying lymph node swelling with hyperemia and renal medullary hemorrhage, also exhibited symptoms not seen in the other pigs, namely trichocardia and fibrin exudation in the lung.

Tissue samples, including from the heart, liver, spleen, lung, kidney, tonsil, lymph nodes, stomach, intestines, brain, and pancreas, were collected from all pigs for quantifying the number of viral DNA copies via qPCR. Viral DNA was detectable in all collected samples except the pancreas, with higher viral loads being detected in the liver, spleen, lung, kidney, tonsil, and lymph nodes (Figure 6D).

No significant difference in tissue viral loads was observed between Group IM and Group ON. Namely, the infection route had no significant effect on the distribution of the virus in tissues. In Group IM, the quantity of viral DNA was the highest in the inguinal lymph nodes, reaching 10 ^8.24 (±0.21)^ copies/g (Figure 6A), while, in Group ON, it was the highest in the spleen, reaching 10 ^7.40 (±1.29)^ copies/g (Figure 6A).

In general, the intramuscular inoculation route resulted in the most consistent gross lesions. The gross lesions of the dead pigs in Group ON were slightly milder than those in Group IM. The mildest gross lesions, except in the heart, lungs, and kidneys, were observed in ON-1, and the viral loads in tissues other than the heart were lower than those in the five dead pigs.

### 3.6. Histopathological Evaluations and Viral Antigen Distribution

Tissue samples from IM-2, ON-2 and ON-1 were selected for additional histopathological evaluations and immunohistochemical studies focused on determining viral antigen distribution. Samples were collected from the heart, liver, spleen, lung, kidney, tonsil, and MLN. The severity of pathological changes was consistent with that of the gross lesions.

In IM-2, as shown in Figure 7A, multifocal hyperemia and mild edema were observed in the heart. The liver displayed connective tissue proliferation, dividing it into numerous false lobules, accompanied by diffuse swelling, spotty hepatocyte necrosis, hyperemia, and macrophage infiltration in the hepatic sinuses, as well as severe edema and infiltration of a small number of macrophages and lymphocytes in the portal area. In the spleen, the boundaries between the red and white pulp were unclear, with severe lymphocytic depletion in the white pulp and extensive engorgement. The lymphocytes were severely necrotic and reduced in number, accompanied by macrophage and lymphocyte infiltration in the white pulp. In the lung, diffuse severe hyperemia and thickening of the alveolar walls were observed, along with alveolar type II epithelial cell hyperplasia, infiltration of a small number of macrophages and lymphocytes in the alveolar walls, and the presence of a small number of red blood cells in the alveolar spaces. The kidney exhibited histopathological changes characterized by multifocal hemorrhage, infiltration of a small number of macrophages, lymphocytes, and neutrophils into hemorrhagic foci, as well as diffuse necrosis of renal tubular epithelial cells. The tonsil showed severe hyperemia and mild edema, accompanied by severe necrosis and detachment of epithelial cells in the crypts. In the MLN, diffuse severe hemorrhage and diffuse lymphocytic necrosis were observed with decreased lymphocyte numbers. The hemorrhagic and necrotic foci exhibited infiltration by macrophages, lymphocytes, and neutrophils.

In Group ON, as shown in Figure 7A, the types of pathological changes in major tissues and organs were similar between IM-2 and ON-2, with differences being primarily seen in the degree of severity. ON-2 exhibited slightly milder lesions than IM-2, with distinct histopathological differences characterized by small hemorrhagic foci in the myocardium, milder hyperemia of the alveolar wall, focal hemorrhage in the subcapsular cortex of the kidney, severe hemorrhage and marked hemosiderin deposition in the subcapsular sinus, and marked edema in the capsule of the MLN.

As shown in Figure 7A, compared to IM-2 and ON-2, most tissues and organs in ON-1 exhibited different types of pathological changes, except for the kidney. The pathological changes in the kidney of ON-1 were similar but milder. Additionally, no obvious pathological changes were observed in the spleen and tonsil of ON-1. However, significant alterations were observed in the heart, including marked hyperemia, edema, and thickening of the epicardium, accompanied by small hemorrhagic foci, cellular debris, and fibrin deposits. Infiltration of macrophages, lymphocytes, a small number of plasma cells, and neutrophils was also noted. The liver showed the mildest histopathological changes, characterized by diffuse hepatocyte swelling and infiltration of a small number of macrophages and lymphocytes in the portal area. In the lung, ON-1 displayed diffuse vascular dilation, pronounced hyperemia, edema, and extreme thickening of the pulmonary pleura, along with small hemorrhagic foci, cellular debris, and fibrin deposits. Macrophages, lymphocytes, and a small number of neutrophils were also present in the pulmonary pleura and alveolar walls. Additionally, hyperemia was observed in the capillaries of the alveolar walls along with alveolar epithelial cell hyperplasia. Marked hemosiderin deposition was also visible in the subcapsular sinuses of the MLN.

The presence of immunolabeled cells for the ASF-specific antigen p30/CP204L was not detected in the hearts or kidneys of the three selected pigs (Figure 7B). The immunolabeled cells in other tissues were primarily macrophages (Figure 7B).

In IM-2, as shown in Figure 7B, viral antigens were detected in the liver, spleen, lung, tonsil, and MLN, with a particularly massive presence of immunolabeled macrophages in the spleen. In ON-2, the detection of viral antigen was consistent with that of IM-2. However, in ON-1, no immunoreactive cells were observed in any of the selected tissues, except for the spleen, which showed a low presence of immunoreactive cells.

## 4. Discussion

This study successfully isolated the ASFV strain HB31A, which shares high genomic similarity with the epidemic virulent Pig/HLJ/18 strain in China, differing by just nine base mutations [23]. Pigs were inoculated with HB31A using both intramuscular and oronasal routes to investigate viral dynamics and pathogenicity.

The incubation periods, disease signs, death times, and gross lesions were consistent among all pigs in Group IM, suggesting that the intramuscular inoculation route led to more reproducible clinical disease. Additionally, viremia was detected two days earlier in Group IM compared to Group ON. This could be due to the virus entering the bloodstream directly, allowing rapid contact with monocytes and macrophages, the target cells of ASFV, which leads to swift disease progression. In contrast, oronasal inoculation involves the virus interacting with mucosal surfaces, triggering an innate immune response before infecting lymph nodes and, subsequently, monocytes and macrophages [20,31,32]. Therefore, the oronasal inoculation route resulted in longer incubation periods, delayed detection of viral DNA in swabs, and later onset of viremia compared to the intramuscular inoculation route.

Rapid diagnosis and culling of infected pigs are essential to curb the spread of ASFV on farms. In Group IM, viral loads were higher and detected earlier in swabs compared to those in Group ON. In both groups, viral shedding was detected first in nasal swabs, followed by oral and rectal swabs. Among these, nasal swabs consistently showed the highest viral loads. However, in the early stages of ASF, viral DNA was not always detectable in nasal swabs. For example, no viral DNA was found in IM-3 on day 2, ON-3 on day 4, and ON-1 on day 5. Given that oral swabs are easy to collect, it is more effective to test both oral and nasal swabs simultaneously during the early stages of ASFV infection.

All pigs in Group IM died within 6.67 (±0.47) days, whereas, in Group ON, ON-1 survived the infection and the other two pigs died at 8 and 10 days, respectively. Although the histopathologic lesion scores were higher in Group IM than in Group ON, the difference between the two groups was not statistically significant. All pigs that succumbed to ASFV infection exhibited clinical symptoms and pathological changes consistent with acute ASF. The clinical symptoms included high fever, inappetence, and lethargy, while the pathological changes included hepatomegaly, splenic and lymph node swelling/hyperemia, petechiae in the renal cortex, and renal medullary hemorrhage.

ON-1 exhibited chronic disease and persistent infection. Necropsy and histopathological evaluations revealed multiple organ injuries, with significant damage to the heart, lung, and kidney. The heart lesions were primarily characterized by trichocardia, while the lungs showed severe fibrinous exudation, explaining the pig’s continued labored breathing until the end of the observation period. Although ON-1 survived the infection with the lowest tissue lesion scores and viral loads, it intermittently excreted the virus and developed low-level viremia. This phenomenon indicates that pigs infected with ASFV in the field can develop non-lethal, chronic disease and persistent infection, with intermittent viral excretion, even when infected with a highly virulent strain. This finding suggests that the survival of ASFV-infected pigs in the field poses a significant threat to uninfected pigs and can lead to severe economic losses in the pig industry.

## 5. Conclusions

This study involved the successful isolation of the HB31A strain from a pig serum sample, followed by its characterization through HAD assay, IFA, and electron microscopy. Animal studies demonstrated that the HB31A strain is highly virulent in pigs via intramuscular and oronasal inoculations. There were no significant differences in tissue viral loads, gross lesions, and pathological changes between the dead pigs in Group IM and Group ON. Pigs in Group IM exhibited more reproducible clinical disease, with a high consistency in disease outcomes. In contrast, pigs in Group ON had a longer incubation period, a longer course of disease, and a later onset of viral shedding.

## Figures and Tables

**Figure 1 vetsci-11-00403-f001:**
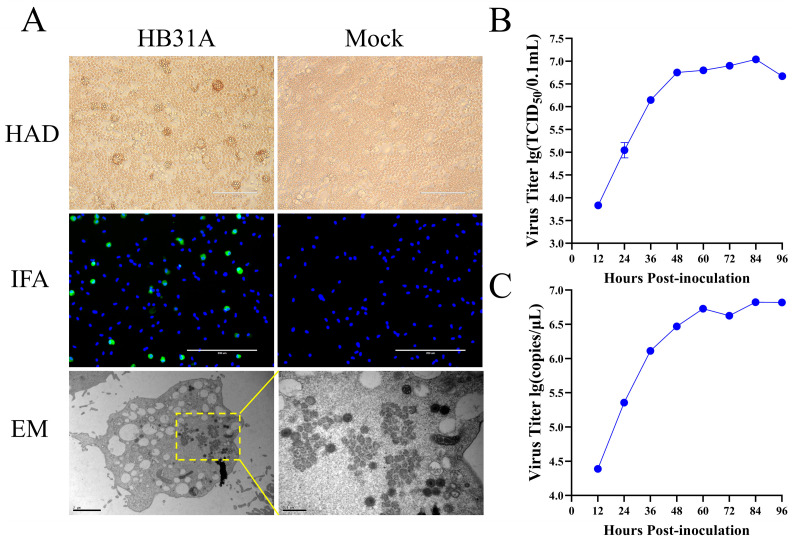
Characterization of HB31A in vitro. (**A**) HAD assay was conducted with HB31A in PAMs. Infection of PAMs using HB31A was performed at an MOI of 0.1, fixed and analyzed by IFA at 48 h p.i. Infection of PAMs using HB31A was performed in T75 cell culture flasks (MOI = 0.1), and the cell pellets were harvested for morphological assessment using an EM. (**B**,**C**) Viral growth curves of HB31A in PAMs. Infection of PAMs using HB31A was performed at an MOI of 0.1, the cell cultures were collected, the virus titers of ASFV were presented as TCID_50_ calculated by the Reed and Muench method (**B**), and viral B646L gene copy numbers were quantified by qPCR (**C**).

**Figure 2 vetsci-11-00403-f002:**
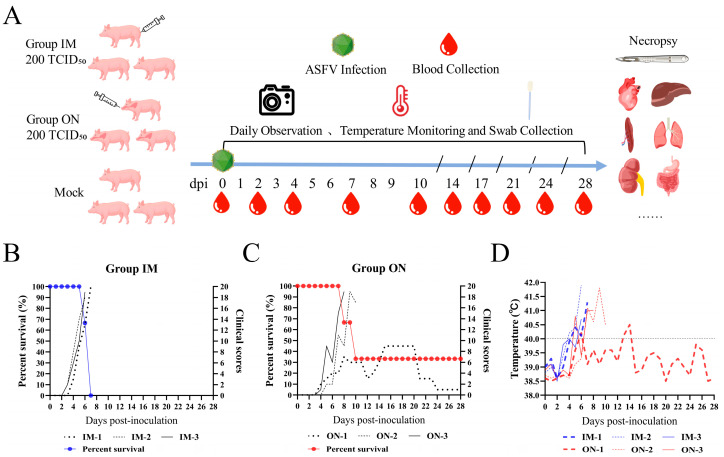
Schematic overview of the animal experiment. Pigs were infected with HB31A, experiencing changes in survival rate, clinical scores, and body temperature. (**A**) Pigs were infected with HB31A via intramuscular and oronasal inoculations. Swabs and blood were collected at the indicated time points, and body temperature changes and disease symptoms were monitored daily after infection. (**B**,**C**) Differences in the survival rate and the clinical scores of pigs infected with HB31A via intramuscular and oronasal inoculations. (**D**) Body temperature changes in pigs infected with HB31A via intramuscular and oronasal inoculations.

**Figure 3 vetsci-11-00403-f003:**
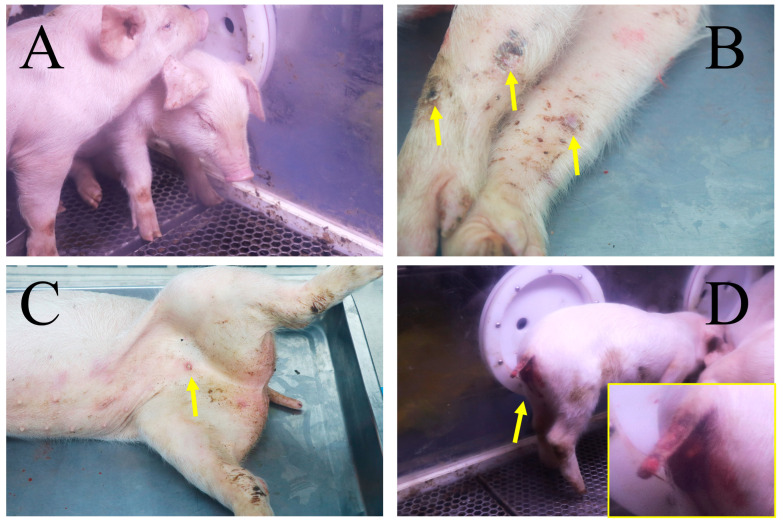
Disease symptoms in pigs following infection with HB31A. Disease symptoms included depression (**A**) and cutaneous necrosis (yellow arrows) (**B**) in both IM and ON groups, papules (yellow arrow) (**C**) in Group IM, and archorrhagia (**D**) in Group ON. The area indicated by the yellow arrow is shown in greater detail in the insert.

**Figure 4 vetsci-11-00403-f004:**
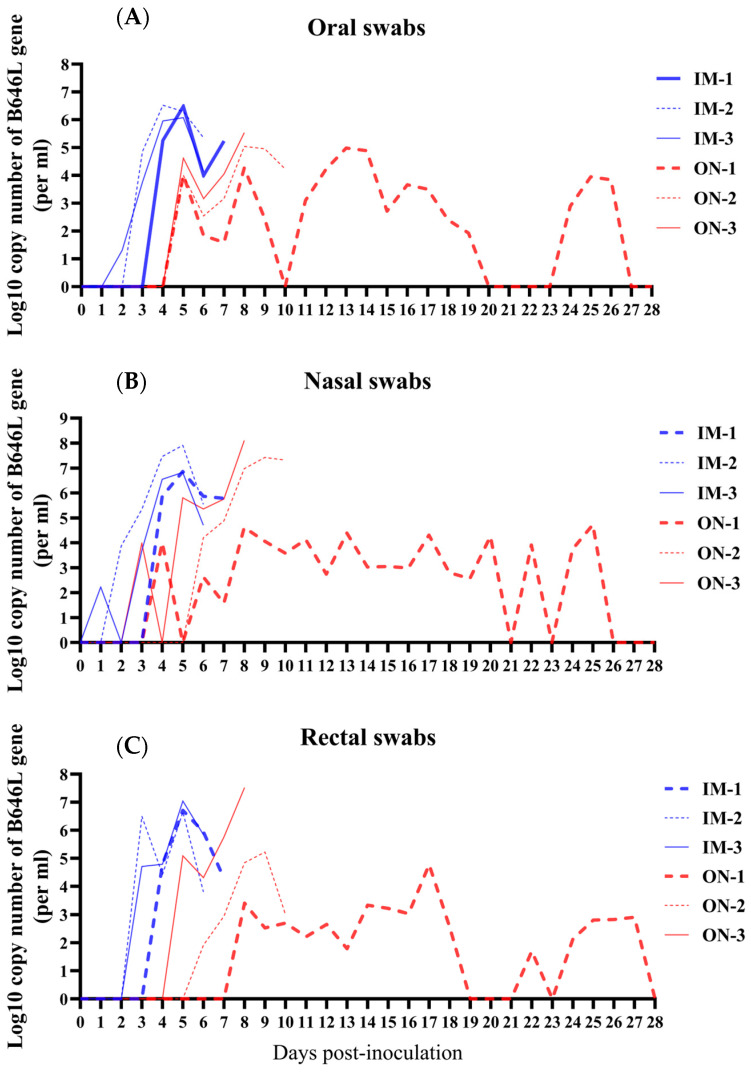
Viral DNA copies were detected from the swabs of pigs infected with HB31A via intramuscular and oronasal inoculations. Viral DNA copies were detected from the oral (**A**), nasal (**B**), and rectal swabs (**C**) of inoculated pigs every day after infection.

**Figure 5 vetsci-11-00403-f005:**
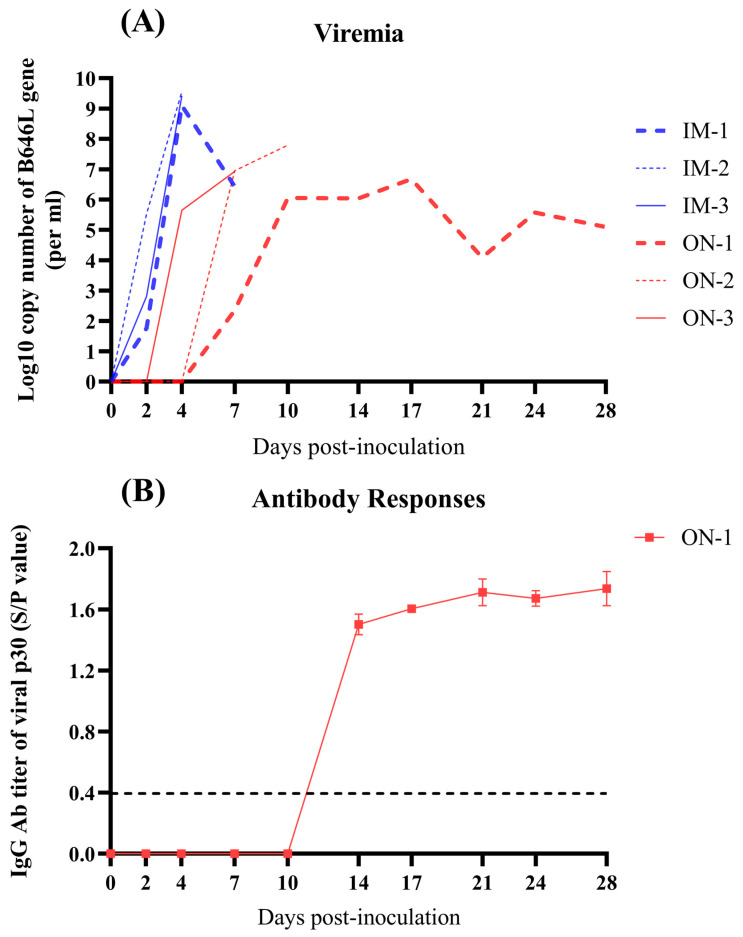
Detection of viral DNA copies in the blood and serum of inoculated pigs at the indicated time points after infection via intramuscular and oronasal inoculations. (**A**) Viral DNA copies in the blood of inoculated pigs. (**B**) Serum of infected pigs was tested for ASFV-specific IgG antibodies targeting the p30 protein, and seroconversion occurred only in ON-1 of Group ON at 14 dpi. The black dotted line indicates the seropositivity with S/P value higher than 0.398.

**Figure 6 vetsci-11-00403-f006:**
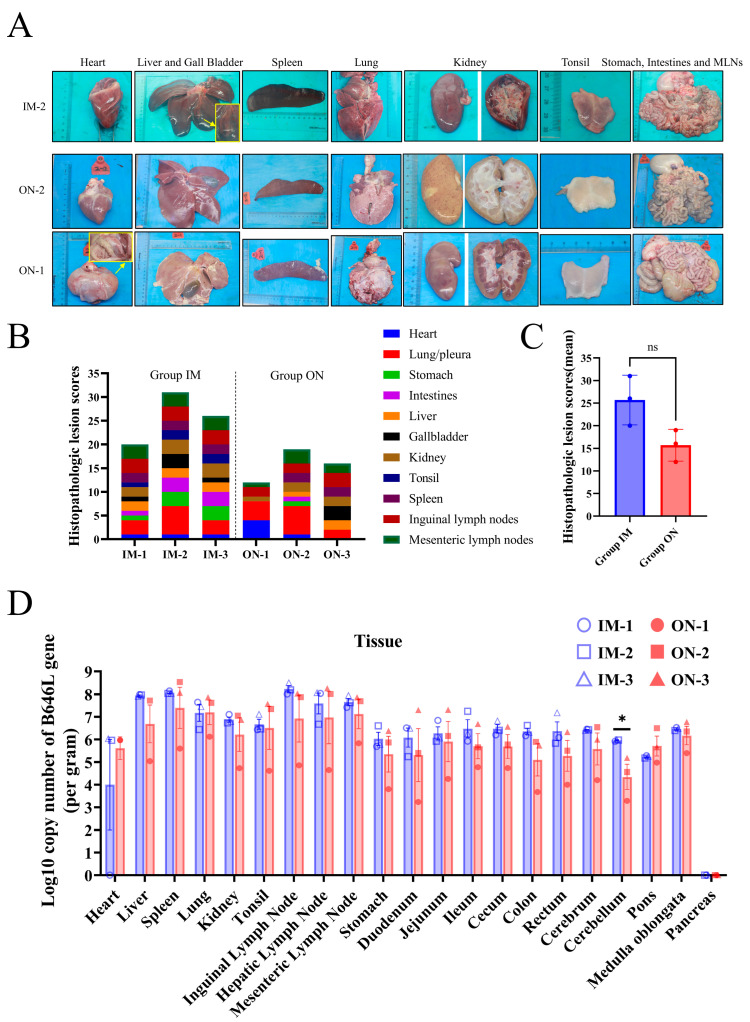
The gross lesions observed in IM-2, ON-2, and ON-1; histopathologic lesion scores and quantification of viral DNA copies in tissues. (**A**) The lesions on the heart, liver, gall bladder, spleen, lung, kidney, stomach, intestines, and mesenteric lymph nodes (MLN) of the pigs with the most severe gross lesions within each experimental group and ON-1 which remained alive after the infection. (**B**,**C**) Histopathologic lesion scores of all pigs in the IM and ON groups. (**D**) The indicated tissue samples were collected from the five dead pigs and ON-1 euthanized at 28 dpi to detect viral DNA copies. “ns” indicates that the values between Group IM and Group ON are not significantly different. * indicates that the values between Group IM and Group ON are significantly different.

**Figure 7 vetsci-11-00403-f007:**
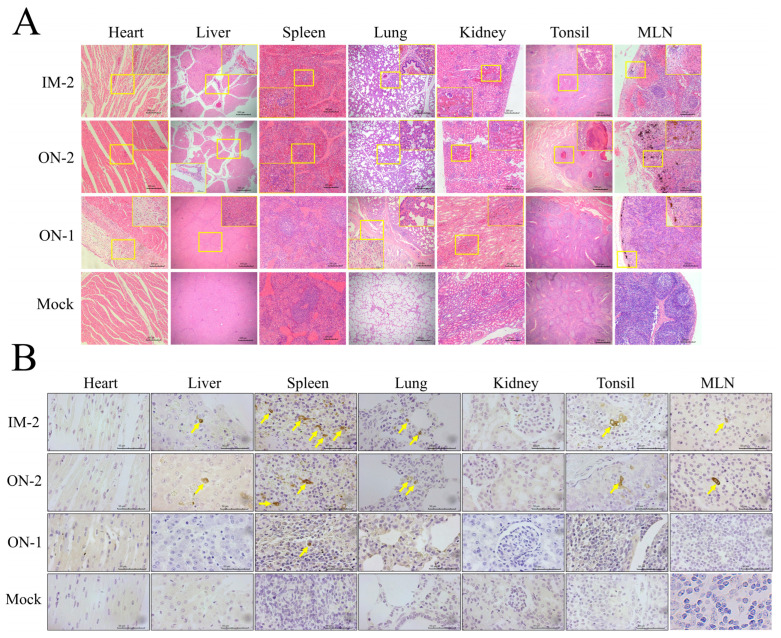
Histopathological lesions observed in IM-2, ON-2 and ON-1. (**A**) Hematoxylin and eosin staining assay; the framed region is shown in greater detail in the insert. MLN, mesenteric lymph node. (**B**) Immunohistochemistry assay. The brown cells are immunolabeled cells and are indicated by yellow arrows.

**Table 1 vetsci-11-00403-t001:** Disease symptoms observed in pigs following inoculation with HB31A.

Earliest Appearance of Disease Signs (dpi)	Pig No. in Group IM			Pig No. in Group ON		
	IM-1	IM-2	IM-3	ON-1	ON-2	ON-3
Fever (≥40 °C)	5	4	4	6	7	5
Inappetence	4	3	3	5	5	4
Lethargy	5	3	3	4	5	5
Wheezing/coughing	/ ^a^	/	/	7	/	/
Nosebleed	/	/	/	/	9	/
Papule	4	/	4	/	/	/
Cutaneous necrosis	/	/	5	8	/	/
Diarrhea	6	4	4	4	/	/
Archorrhagia	/	/	/	/	/	8
Death	7	6	7	NA ^b^	10	8

^a^ No manifestations of disease symptoms. ^b^ The pig remained alive after the infection.

**Table 2 vetsci-11-00403-t002:** Incubation period and virological parameters in pigs infected with HB31A.

Group	Incubation period (±SD)	Oral Shedding		Nasal Shedding		Rectal Shedding		Viremia
		Days to the Onset (±SD)	Maximum Titer (±SD) ^a^	Days to the Onset (±SD)	Maximum Titer (±SD) ^a^	Days to the Onset (±SD)	Maximum Titer (±SD) ^a^	Maximum Titer (±SD) ^a^
IM	4.33 (±0.47) ^b^	3.00 (±0.82) ^c^	6.36 (±0.20) ^d^	2.33 (±1.25) ^b^	7.20 (±0.50) ^b^	3.33 (±0.47) ^c^	6.80 (±0.17) ^b^	9.36 (±0.19) ^d^
ON	6.00 (±0.82)	5.00 (±0)	5.19 (±0.25)	4.33 (±1.25)	6.75 (±1.46)	6.33 (±1.25)	5.84 (±1.20)	7.13 (±0.48)

^a^ Log10 copies/mL. ^b^ The values between Group IM and Group ON are not significantly different (*p* > 0.05). ^c^ The values between Group IM and Group ON are significantly different (*p* ≤ 0.05). ^d^ The values between Group IM and Group ON are very significantly different (*p* ≤ 0.01).

## Data Availability

Data are contained within the article and Appendix A.

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
