# Peer review of "In Vivo Study of Inoculation Approaches and Pathogenicity in African Swine Fever"

_vetsci, 2024, doi:10.3390/vetsci11090403_

Round 1
Reviewer 1 Report
Comments and Suggestions for Authors
It is an interesting article in which the author describes the identification and pathogenesis of a recently isolated strain of ASFV in pigs.
Here are some comments:
1- In the method section: You used the method (including primers) of King et al., 2003 for qPCR for the genotype 2 strain that you used in the current article. King et al., didn't use any genotype 2 in their paper, though it may still work. It would be nice if you at least add the primer/probe sequence that you have used in the current study.
2- In vivo experimental design: My only comment would be about the number of animals in each group, do you think 3 animals is a sufficient number, especially for the ON group?
3- The quality of the image in Fig 1 is of question. I can't see the scale bar details.
4- For virus growth dynamics, it would be nice to see the cell lysate virus titer vs supernatant. I was wondering if you had a chance to assess the infectious virus and or DNA copies in the cell lysate as well?
5- Very interesting to see that IM group showed viral shedding in nasal swabs as early as 1DPI (earlier than its presence in the blood) and not until 3 DPI in the ON group and that makes me wonder how you administer the viral dosage in ON group have you used a nebulizer, did you have to stress the animal before the infection for a successful challenge...etc.
6- In Fig 4: you need to denote that the * indicates significance.
7- In Fig 6: Was very hard for me to see some lesions. Also in the MOCK one, the heart looks congested, unless the state of exsanguination for this Mock animal was incomplete which makes the heart look congested like this. You may need to work on this figure or just take it away.
Comments on the Quality of English Language
Minor to moderate English editing and font adjustment may be needed
Author Response
Reviewers’ comments:
Reviewer #1:
Comment 1: In the method section: You used the method (including primers) of King et al., 2003 for qPCR for the genotype 2 strain that you used in the current article. King et al., didn't use any genotype 2 in their paper, though it may still work. It would be nice if you at least add the primer/probe sequence in the current study.
Response 1: Thank you for your advice. We have added the sequence information in Materials and Methods and provided the complete procedure in Supplementary Material 1. The forward primer used was 5’-CTG-CTC-ATG-GTA-TCA-ATC-TTA-TCG-A-3’, the reverse primer was 5’-GAT-ACC-ACA-AGA-TC(AG)-GCC-GT-3’, and the probe was 5’-FAM-CCA-CGG-GAG-GAA-TAC-CAA-CCC-AGT-G-3’ TAMRA. The target amplification gene region is very stable, and this primer/probe sequence is also commonly used in many published articles about genotype II ASFV strain, such as:
- Zhao, Dongming et al. “Replication and virulence in pigs of the first African swine fever virus isolated in China.” Emerging microbes & infections vol. 8,1 (2019): 438-447. doi:10.1080/22221751.2019.1590128;
- Sun, Encheng et al. “Emergence and prevalence of naturally occurring lower virulent African swine fever viruses in domestic pigs in China in 2020.” Science China. Life sciences vol. 64,5 (2021): 752-765. doi:10.1007/s11427-021-1904-4;
- Zhao, Dongming et al. “Highly lethal genotype I and II recombinant African swine fever viruses detected in pigs.” Nature communications vol. 14,1 3096. 29 May. 2023, doi:10.1038/s41467-023-38868-w.
Comment 2: In vivo experimental design: My only comment would be about the number of animals in each group, do you think 3 animals is a sufficient number, especially for the ON group?
Response 2: Thank you for your constructive comment. We agree that the optimal number of pigs per group should be five. However, due to limited conditions, we are conducting animal experiments in a negative-pressure isolator of a small ABSL-3 laboratory. The space in the negative-pressure isolator is too limited to accommodate five pigs, and the air is also insufficient for five pigs to breathe.
Comment 3: The quality of the image in Fig 1 is of question. I can't see the scale bar details.
Response 3: Fig 1 in the word document was compressed, and we have replaced them with high-definition images.
Comment 4: For virus growth dynamics, it would be nice to see the cell lysate virus titer vs supernatant. I was wondering if you had a chance to assess the infectious virus and or DNA copies in the cell lysate as well?
Response 4: Thank you for your question. In our research, we did not compare the virus titer between the supernatant and the cell lysate. We employed a method of three freeze-thaw cycles of the cell culture to ensure that the virus was fully released from the cell debris. We then centrifuged the sample to remove the cell debris and used the supernatant for measuring viral growth. To avoid any potential misunderstanding by readers, we have provided a more detailed description of the materials and methods related to the viral growth dynamics in the article.
Comment 5: Very interesting to see that IM group showed viral shedding in nasal swabs as early as 1DPI (earlier than its presence in the blood) and not until 3 DPI in the ON group and that makes me wonder how you administer the viral dosage in ON group have you used a nebulizer, did you have to stress the animal before the infection for a successful challenge...etc.
Response 5: It cannot be said that IM group showed viral shedding earlier in nasal swabs than its presence in the blood. To avoid negatively impacting the health of the Twenty-day-old pigs due to frequent blood sampling, we collected blood samples at 0, 2, 4, 7, 10, 14, 17, 21, 24, and 28 dpi, and did not collect blood samples at 1dpi.
Before the infection, the animals were not subjected to any stress. Instead of using a nebulizer, a 1 ml syringe without a needle was used for inoculation. The pigs in the two experimental groups were inoculated with HB31A at a dose of 200 TCID50 (in 200 μL of cell culture medium) via intramuscular inoculation (IM) or oronasal inoculation (ON). Specifically, pigs in Group ON were inoculated with 100 TCID50 orally and nasally each.
Comment 6: In Fig 4: you need to denote that the * indicates significance.
Response 6: To better compare the viral shedding time between the two groups of pigs, we analyzed the data at the individual level and adjusted the figure style. The new figure there is no * present in the new figure.
Comment 7: In Fig 6: Was very hard for me to see some lesions. Also in the MOCK one, the heart looks congested, unless the state of exsanguination for this Mock animal was incomplete which makes the heart look congested like this. You may need to work on this figure or just take it away.
Response 7: Fig 6 in the word document was compressed, and we have replaced them with high-definition images and take the mock picture away.
Comments on the Quality of English Language:Minor to moderate English editing and font adjustment may be needed.
Response:We have improved the writing of the article and adjusted the font.
Reviewer 2 Report
Comments and Suggestions for Authors
In general, the aim of the work is to isolate a viral strain of ASF, characterise it and verify its pathogenicity and/or virulence by means of an experimental infection. For this, an experimental infection was carried out using two different inoculation routes, thus providing the opportunity to compare the different disease course and epidemiological dynamics.
The work therefore begins by describing the isolation of the viral strain in line with the methods in use throughout the world; if anything, it adds observation under an electron microscope, which adds nothing to the other, much more effective methods; beyond the curious aspect of knowing that observation under an electron microscope was successful, it is not clear the purpose of this finding and above all what the point is of including it in this study.
The best part is the Methods Materials: these are described in an appropriate form. In any case, there is no clear method of comparing the two groups of pigs used in the experimental infection.
The results are described in a redundant form and the authors often insert commentary sentences that should find a place in the discussion.
The worst chapter is that of the discussion, which should be completely rewritten; the authors lose sight of the objectives of the work and get lost in very banal comments. They repeat observations that are well known to all and do not dwell on the findings of the study. The most interesting part would be to compare the observations obtained from the two groups of pigs infected by the OroNasal or IntraMuscular route. In reality, the authors have not applied a correct method to make an effective comparison. In practice, they merely listed the clinical trend and the list of laboratory test findings in the results, but in a sterile form.
In particular, they should have applied a quantitative method for clinical symptoms (clinical score) and for necropsy lesions (pathological score) and thus compared the results more incisively.
The work should therefore be revised by applying a major review.
Author Response
Reviewer #2:
In general, the aim of the work is to isolate a viral strain of ASF, characterise it and verify its pathogenicity and/or virulence by means of an experimental infection. For this, an experimental infection was carried out using two different inoculation routes, thus providing the opportunity to compare the different disease course and epidemiological dynamics.
The work therefore begins by describing the isolation of the viral strain in line with the methods in use throughout the world; if anything, it adds observation under an electron microscope, which adds nothing to the other, much more effective methods; beyond the curious aspect of knowing that observation under an electron microscope was successful, it is not clear the purpose of this finding and above all what the point is of including it in this study.
The best part is the Methods Materials: these are described in an appropriate form. In any case, there is no clear method of comparing the two groups of pigs used in the experimental infection. The results are described in a redundant form and the authors often insert commentary sentences that should find a place in the discussion.
The worst chapter is that of the discussion, which should be completely rewritten; the authors lose sight of the objectives of the work and get lost in very banal comments. They repeat observations that are well known to all and do not dwell on the findings of the study. The most interesting part would be to compare the observations obtained from the two groups of pigs infected by the OroNasal or IntraMuscular route. In reality, the authors have not applied a correct method to make an effective comparison. In practice, they merely listed the clinical trend and the list of laboratory test findings in the results, but in a sterile form.
In particular, they should have applied a quantitative method for clinical symptoms (clinical score) and for necropsy lesions (pathological score) and thus compared the results more incisively.
The work should therefore be revised by applying a major review.
Thank you for your suggestions. You pointed out many shortcomings in the article, and based on your advice, I made the corresponding revisions. I will respond to each of your points accordingly.
Response 1:Electron microscopy is one of the methods used to characterize successful virus isolation. Many articles on African swine fever virus isolation have employed electron microscopy, which is why we included it in our paper. References include:
- Zhao, Dongming et al. “Replication and virulence in pigs of the first African swine fever virus isolated in China.” Emerging microbes & infections vol. 8,1 (2019): 438-447. doi:10.1080/22221751.2019.1590128;
- Sun, Encheng et al. “Emergence and prevalence of naturally occurring lower virulent African swine fever viruses in domestic pigs in China in 2020.” Science China. Life sciences vol. 64,5 (2021): 752-765. doi:10.1007/s11427-021-1904-4;
- Sun, Encheng et al. “Genotype I African swine fever viruses emerged in domestic pigs in China and caused chronic infection.” Emerging microbes & infections vol. 10,1 (2021): 2183-2193. doi:10.1080/22221751.2021.1999779.
Response 2:We have added content on clinical scoring, the virus shedding time in oral, nasal, and rectal swabs, as well as the maximum titer and pathological scoring in the paper. We also used statistical methods to perform significance analysis between the two groups, allowing for a more effective comparison of the data from the two groups of pigs.
Response 3:We edited the results section to make it more concise, avoided redundant and monotonous descriptions. Additionally, some commentary sentences were removed from the results section and placed in the discussion section.
Response 4:We revised the discussion section by removing content that was not closely related to the research objectives and focused on discussing and analyzing the data from the two groups of pigs.
Once again, thank you for your feedback; it has been incredibly helpful for my paper.

Reviewer 3 Report
Comments and Suggestions for Authors
Title of reviewed article:
Isolation, identification and pathogenicity of African swine fever virus through intramuscular and oronasal inoculations
Veterinary sciences
Comments
Relevance of the title to the content of the manuscript
If the title can be more concise for instance, “In vivo study of inoculation approaches and pathogenicity in African Swine Fever”
Abstract
Line 25: What type of damage does the author meant? Pathological or financial, if either or both, please state accordingly.
Line 33 to 35: What is the time period of shedding of disease? Please state accordingly.
The authors state the aim of the study is “…To investigate the pathogenicity of HB31A, we performed animal experiments on twenty-day-old pigs through intramuscular and oronasal inoculations”. The investigators conducted an in vivo study of the African swine fever (ASF) virus in swine research animals. Intramuscular and oronasal inoculations are methods of study. It is recommended that these details be included in the Methods section so that the abstract section is more concise.
Approach or methodology
Lines 77 to 80: “HB31A was used to inoculate pigs via intramuscular and oronasal inoculations…”. It should be written in coherence as the Method that either intramuscular “or” oronasal inoculators are conducted instead of “and”. This is because authors stated in introduction (Lines 59 to 67) that intramuscular inoculation has weakness and this experiment is to provide an investigator or an alternative.
Lines 77 to 80: The authors stated 10 metrics in its outcome measures. Please write the metrics separately from the sentence that discussed inoculation approaches. For example, one can write: “…The outcome of inoculation is measured as pathogenicity (define ), survival rate (percentage or ratio), incubation periods (days), (clinical signs) disease signs, viral shedding, viremia, antibody responses, viral loads in various tissues, gross lesions, and histopathological changes…”
Can authors please describe more clearly what does it mean by inoculation intraoropharyngeally and intranasopharyngeally? Is the virus load injected subcutaneously, intramuscularly or any other approaches?
Can Authors please clarify observation period in the Method section? That is, how long was the observation period? The Figures in A to C appears to indicate 30 days. If so, please state in Method section and why the 30-day period is chosen as end date? Authors indicate days of inoculation, latent date are important in terms of culling. Could authors please state the time decision framework.
Histopathology in subsection 3.6
Authors please state whether a human or veterinary pathologist, certified or specialist reads the histopathology slide so the description of pathology results are communicated in context.
Institutional Review Board Statement
This is an animal study manuscript. Please state animal research ethical clearance. Mainly, are these research pigs and is there any animal research ethical clearance?
Tables and Figures
Table 1: Please organize the fonts and widths for a whole word to be adjusted in each column.
Writing and grammatical comments
Fragmented sentence line 75 and 76: “And the isolate was characterized by HAD assay, immunofluorescence assay, electron microscopy”. Please re-write.
Line 96: Please put full stop after the citation, not before and after. Please check the rest of the writing for similar issues.
Line 185: Full stop to be placed after (B, C).
Lines 185 to 155: does (B, C) mean Table B and Table C? If so, please label accordingly.
Comments on the Quality of English Language
There are minimum connections between sentences. This renders the writing with some fragmentation. Punctuations in certain places of the writing requires editing.
Author Response
Reviewer #3:
Comment 1: If the title can be more concise for instance, “In vivo study of inoculation approaches and pathogenicity in African Swine Fever”
Response 1:Thank you for your suggestion, indeed, the proposed title is better, we have revised the title accordingly.
Comment 2: Line 25: What type of damage does the author meant? Pathological or financial, if either or both, please state accordingly.
Response 2:Thank you for your question. What we intended to convey was the pathological and financial damage. But to make the summary more concise, we removed this sentence.
Comment 3: Line 33 to 35: What is the time period of shedding of disease? Please state accordingly.
Response 3:Thank you for your valuable comment. Actually, we detected the virus as early as 1-3 dpi in Group IM and 3-5 dpi in Group ON. But unfortunately, we were unable to estimate the exact shedding period, as some animals shed the virus intermittently, as illustrated in Figure 4.
Comment 4: The authors state the aim of the study is “…To investigate the pathogenicity of HB31A, we performed animal experiments on twenty-day-old pigs through intramuscular and oronasal inoculations”. The investigators conducted an in vivo study of the African swine fever (ASF) virus in swine research animals. Intramuscular and oronasal inoculations are methods of study. It is recommended that these details be included in the Methods section so that the abstract section is more concise.
Response 4:We completely agree with your comment. We edited this sentence accordingly.
Comment 5: Lines 77 to 80: “HB31A was used to inoculate pigs via intramuscular and oronasal inoculations…”. It should be written in coherence as the Method that either intramuscular “or” oronasal inoculators are conducted instead of “and”. This is because authors stated in introduction (Lines 59 to 67) that intramuscular inoculation has weakness and this experiment is to provide an investigator or an alternative.
Response 5:Thank you for your suggestion, we edited this sentence accordingly.
Comment 6: Lines 77 to 80: The authors stated 10 metrics in its outcome measures. Please write the metrics separately from the sentence that discussed inoculation approaches. For example, one can write: “…The outcome of inoculation is measured as pathogenicity (define), survival rate (percentage or ratio), incubation periods (days), (clinical signs) disease signs, viral shedding, viremia, antibody responses, viral loads in various tissues, gross lesions, and histopathological changes…”
Response 6:We edited this sentence accordingly.
Comment 7: Can authors please describe more clearly what does it mean by inoculation intraoropharyngeally and intranasopharyngeally? Is the virus load injected subcutaneously, intramuscularly or any other approaches?
Response 7:We edited this sentence accordingly. The pigs in the two experimental groups were inoculated with HB31A at a dose of 200 TCID50 (in 200 μL of cell culture medium) via intramuscular inoculation (IM) or oronasal inoculation (ON). Specifically, pigs in Group ON were inoculated with 100 TCID50 orally and nasally each.
Comment 8: Can Authors please clarify observation period in the Method section? That is, how long was the observation period? The Figures in A to C appears to indicate 30 days. If so, please state in Method section and why the 30-day period is chosen as end date? Authors indicate days of inoculation, latent date are important in terms of culling. Could authors please state the time decision framework.
Response 8:Thank you for your suggestion, we have already stated in the Materials and Methods section that the observation period is 28 days. The image also shows 28 days. The reason we chose 28 days as the observation period is that many related studies have used this duration and also because the experiment was done in ABSL-3 lab, we can’t observe it in a longer period. References include:
- Sun, Encheng et al. “Genotype I African swine fever viruses emerged in domestic pigs in China and caused chronic infection.” Emerging microbes & infections vol. 10,1 (2021): 2183-2193. doi:10.1080/22221751.2021.1999779.
- Zhao, Dongming et al. “Highly lethal genotype I and II recombinant African swine fever viruses detected in pigs.” Nature communications vol. 14,1 3096. 29 May. 2023, doi:10.1038/s41467-023-38868-w
- Sun, Encheng et al. “Emergence and prevalence of naturally occurring lower virulent African swine fever viruses in domestic pigs in China in 2020.” Science China. Life sciences vol. 64,5 (2021): 752-765. doi:10.1007/s11427-021-1904-4;
Comment 9: Authors please state whether a human or veterinary pathologist, certified or specialist reads the histopathology slide so the description of pathology results are communicated in context.
Response 9:The histopathological and immunohistochemical analyses were carried out by veterinary pathologists in Wuhan Baiqiandu Biotechnology Co., Ltd. We have added this statement in the materials and methods section.
Comment 10: This is an animal study manucript. Please state animal research ethical clearance. Mainly, are these research pigs and is there any animal research ethical clearance?
Response 10:Thank you for your suggestion, we have added to the Institutional Review Board Statement in our article. All animal-related study procedures were performed according to the Care and Use of Laboratory Animals of the Research Ethics Committee (Ethical code: HZAUSW-2022-0017, approved on 4 September 2022).
Comment 11: Table 1: Please organize the fonts and widths for a whole word to be adjusted in each column.
Response 11:Thank you for your reminder. We have made the corresponding modifications to the table.
Comment 12: Fragmented sentence line 75 and 76: “And the isolate was characterized by HAD assay, immunofluorescence assay, electron microscopy”. Please re-write.
Response 12:Thank you for your suggestion, we edited this sentence accordingly.
Comment 13: Line 96: Please put full stop after the citation, not before and after. Please check the rest of the writing for similar issues.
Response 13:Thank you for your suggestion, we have made the corresponding modifications and reviewed the entire article.
Comment 14: Line 185: Full stop to be placed after (B, C).
Response 14:We have rewritten this part.
Comment 15: Lines 185 to 155: does (B, C) mean Table B and Table C? If so, please label accordingly.
Response 15:We have rewritten this part.
Comment 16: There are minimum connections between sentences. This renders the writing with some fragmentation. Punctuations in certain places of the writing requires editing.
Response 16:Thank you for your suggestions. We have reviewed our article, identified some issues, and made the revisions accordingly.
